Kernel density estimation of allele frequency including undetected alleles

Aoki Satoshi aokis1ll1@gmail.com
http://orcid.org/0000-0002-9563-457X Fukasawa Keita
Biodiversity Division, National Institute for Environmental Studies , Tsukuba, Ibaraki , Japan
Bastolla Ugo
Electronic publication date: 2024 Apr 22
Publication date: 2024
Volume: 12
Electronic Location ID: e17248
Received 2023 Nov 24; Accepted 2024 Mar 25
Copyright: © 2024 Aoki and Fukasawa
Copyright year: 2024
Copyright holder: Aoki and Fukasawa
License: This is an open access article distributed under the terms of the Creative Commons Attribution License, which permits unrestricted use, distribution, reproduction and adaptation in any medium and for any purpose provided that it is properly attributed. For attribution, the original author(s), title, publication source (PeerJ) and either DOI or URL of the article must be cited.
License URL: https://creativecommons.org/licenses/by/4.0/

Keywords: Kernel density estimation, Allele frequency, Genetic diversity

Funding: JSPS KAKENHI JP22J00445 This study was supported by JSPS KAKENHI Grant Number JP22J00445. The funders had no role in study design, data collection and analysis, decision to publish, or preparation of the manuscript.

==============================
Whereas undetected species contribute to estimation of species diversity, undetected alleles have not been used to estimated genetic diversity. Although random sampling guarantees unbiased estimation of allele frequency and genetic diversity measures, using undetected alleles may provide biased but more precise estimators useful for conservation. We newly devised kernel density estimation (KDE) for allele frequency including undetected alleles and tested it in estimation of allele frequency and nucleotide diversity using population generated by coalescent simulation as well as well as real population data. Contrary to expectations, nucleotide diversity estimated by KDE had worse bias and accuracy. Allele frequency estimated by KDE was also worse except when the sample size was small. These might be due to finity of population and/or the curse of dimensionality. In conclusion, KDE of allele frequency does not contribute to genetic diversity estimation.

Introduction

Estimation of genetic diversity is crucial in the fields of evolutionary biology, population genetics and conservation biology. In the estimation of species diversity, inclusion of undetected species contributes to an accurate estimation of diversity (e.g., Chao et al., 2017). However, to the best of our knowledge, undetected alleles have not yet been used to estimate genetic diversity. This may be because random sampling of alleles theoretically guarantees an unbiased estimation of nucleotide diversity or Watterson’s theta (Nei & Tajima, 1981; Watterson, 1975). However, they are not shown as uniformly minimum variance unbiased estimators. This indicates the presence of other estimators with smaller variances, especially if the estimators are not unbiased considering the Cramér-Rao bound. Therefore, biased but more accurate estimators of genetic diversity may be useful for conservation, especially as underestimation of genetic diversity is more permissible than overestimation due to the difficulty in recovering lost genetic diversity. Moreover, accurate estimation of the frequency of undetected alleles can improve the expression of the haplotype network (e.g., Bandelt, Forster & Röhl, 1999). Therefore, we used information on undetected alleles for better estimation of genetic diversity in this study.

Unlike undetected species, the base composition of undetected allele sequences is necessary to estimate genetic diversity. Sequences of detected alleles are related to those of undetected alleles; when one sequence is detected in a population, sequences which are genetically closer to the detected one are more likely to be detected due to potential mutations in the population. Therefore, to identify the sequences of undetected alleles and use phylogenetic relationships to estimate undetected alleles, we employed the kernel density estimation (KDE) (Parzen, 1962). KDE is a method to estimate the probability density function by assuming the potential probability for the range around the detected data (samples) and then summing the probabilities. KDE is also used to estimate the probability mass function (e.g., Wansouwé, Kokonendji & Kolyang, 2015). We found an analogy between ordinal KDE and the estimation of allele frequency, including that of undetected alleles: First, both the allele frequency and probability density function sum up to one, and their portion is always non-negative. Second, the distance between nucleotide sequences can be expressed as a discrete space (nucleotide sequence space) which corresponds to the space where KDE is applied. Therefore, KDE can be used to estimate allele frequency as a probability mass function of nucleotide sequences, and the extent of the kernel function around the data (detected alleles) covers the frequency of undetected alleles. Now, we explain how to use KDE to estimate allele frequency including that of undetected alleles. Then, we explain and discuss our experiments to validate the efficiency of the KDE method.

Theory

KDE of allele frequency

Allele frequency ranges from 0 to 1, and the sum of all frequencies is always 1. Here, we consider the nucleotide sequence space as discrete; the distance between alleles is equivalent to the number of substitutions between them, and we do not consider mixed bases. We assume that all the alleles has the same length, and the nucleotide sequence space has dimensions equal to the length of the alleles. In this case, the allele frequency can be considered as a probability mass function defined on the nucleotide sequence space. Although KDE is usually applied to a continuous probability density function, the allele frequency lies on the discrete nucleotide sequence space. Furthermore, whereas ordinal real-number components of space change bidirectionally (positive and negative), a base as a component of the sequence space can change (mutate) in three directions (bases). Due to this discreteness and particularity of the sequence space, KDE of allele frequency is abnormal but possible.

Choice of the kernel function in KDE does not have a large impact on the result of KDE, but because the KDE of allele frequency is on a unique sequence space as described above, a special kernel function is required. To define the kernel function, we first assume the probability P(g,μ) that one base will be another after g generations (e.g., A = >T). This is calculated from the probability recurrence relation as:

P(g,μ)=1/4+(1/3μ−1/4)(1−4/3μ)g−1,

where μ is the mutation rate per generation per base locus, which is concordant with the Jukes-Cantor substitution model (Jukes & Cantor, 1969). This equation assumes no recombination and no population size changes, but the content of the equation is not important. Then, we consider the probability QP(a,x,P) that one sequence a of length l will be another sequence with x new bases after g generations (e.g., AAA = >ATT when l=3,x=2). Assuming the independence of the mutations, it will be:

QP(a,x,P)={1−3P(g,μ)}l−xP(g,μ)x.

Now we can consider that P(g,μ) and QP(a,x,P) correspond to the bandwidth (smoothing parameter) and kernel function, respectively. This is because summing up QP(a,x,P) for all x equals to 1, and P(g,μ) can be considered as one variable which defines the shape of the kernel function QP(a,x,P). The mutation rate μ and the number of generations g in P(g,μ) are not necessary to conduct KDE because the value of P(g,μ) is selected as one variable through a bandwidth optimization method (described in later section). In this scheme, we assume the sequence length l is fixed. Insertions and deletions must be excluded since they are completely different from substitutions, especially in the point that multiple cites are often inserted or deleted by a single mutation. Using this kernel function, KDE of allele frequency can be expressed by the following equation:

(1) f^(a)=1n∑i=1n⁡∑x=0l⁡QP(ai,x,P),#(1)

where f^(⋅) is the estimated frequency mass function of the allele frequency, a is all alleles, ai is an allele in the sample, n is the sample size and x indicates the number of base changes from ai. Please note that the probabilities given by QP(ai,x,P) are recorded on coordinates or alleles on the nucleotide sequence space. We will discuss the space later.

There are many kinds of methods to optimize the bandwidth (Heidenreich, Schindler & Sperlich, 2013), but we could only implement the least square cross validation (LSCV) method and the likelihood cross validation (LCV) method (Zambom & Dias, 2013). This is because many other methods, such as the plug-in methods, depend on the Taylor expansion for approximation (Zambom & Dias, 2013), but differential of the nucleotide sequence space is difficult to define, and only the first order differential can be considered for the nucleotide sequence space since the maximum distance between bases is one.

Nucleotide sequence space and its compression

KDE calculation is conducted on the nucleotide sequence space. However, the sequence space is generally too large for ordinary computers. For example, the space of 1,000-bp sequences has 41000≈10602 coordinates. Only when sequences have a few bases, KDE can be conducted directly on the sequence space. Otherwise, the sequence space must be compressed.

One way to compress the nucleotide sequence space is to use “distance space”. The distance space assigns one coordinate to each detected allele. Undetected alleles which are not obtained by sampling but estimated by KDE are recorded by their distances from the detected alleles. For example, when the detected alleles are “AA” and “AT”, the total distance space will be (0, 1), (1, 0), (1, 1), (1, 2), (2, 1) and (2, 2), where the first and second components of the coordinates correspond to the distance from the sequence “AA” and “AT”, respectively. Figs. 1 and 2 show the results of KDE in the sequence and distance spaces, respectively, when three “AA” and three “AT” alleles are detected. The size of the sequence space for the detected alleles “AA” and “AT” is 42 whereas the corresponding distance space has only six coordinates as shown above. For example, the coordinate (1, 1) in the distance space corresponds to “AG” and “AC” in the nucleotide sequence space. Thus, the distance space can reduce the space size.

Figure 1 Sample allele frequencies and allele frequencies estimated by KDE on sequence space using the least squares cross validation.

The original samples are three “aa” and three “at” sequences shown as gray bars, and the estimated frequencies are shown as a line graph. The alleles “ag” and “ac” have a little high frequency because they are one distance away from both of the detected alleles, “aa” and “at”. The alleles “gg” to “tc” have the lowest frequency because they are most distant from the detected alleles.

Figure 2 Sample allele frequencies and allele frequencies estimated by KDE on distance space using the least squares cross validation.

The original samples are three “aa” and three “at” sequences shown as gray bars, and the estimated frequencies are shown as a line graph.

However, further compression of the sequence space is practically necessary. For this purpose, maximum range of the kernel function QP(a,x,P) is limited to a given positive integer by changing l in the Eq. (1) to another integer. We refer to this integer as the “mutation number”. Owing to this range limitation, the probability density of undetected alleles farther than the mutation number from a detected allele becomes zero, and such undetected alleles no longer need to be recorded on the distance space. Therefore, all components of the coordinates in the distance space larger than the mutation number can be recorded as infinity and serve for space compression. For example, when the detected alleles are “AAA”, “AAT” and “GGG”, the size of the distance space is 29. If mutation number 1 is applied to this distance space, the size of the space will be seven composed of (0, 1, 3), (1, 0, 3), (3, 3, 0), (1, 1, Inf), (1, Inf, Inf), (Inf, 1, Inf) and (Inf, Inf, 1). Setting a larger mutation number requires more memory and computation time but provides more precise results. However, this improvement in precision decreases as the mutation number increases because the estimated frequency of the undetected alleles decreases as the undetected alleles are farther from a detected allele. On the other hand, setting the mutation number 0 is equivalent to not applying KDE and calculating the allele frequency by counting (cf. gray bars in Figs. 1 and 2). Practically, we suppose that using mutation number of 1 or 2 is sufficient in most cases, but larger mutation number may be desirable when using data with high mutation rates, such as virus sequences.

Approximate nucleotide diversity

Calculation of nucleotide diversity from the result of KDE is not possible when using distance space. Therefore, we devised approximate nucleotide diversity as follows. Nucleotide diversity π is sum of the product of all pairs of allele frequencies and substitution rate between the pair, namely,

π=∑i=1N⁡∑j=1N⁡pipjπij,

where N is the total number of allele types, pi and pj are allele frequency of the sequence i and j, and πij is the substitution rate (Nei & Li, 1979; Nei & Tajima, 1981). The substitution rate is the distance between alleles divided by the sequence length. The frequencies and distances between the detected alleles can be directly obtained from the frequencies estimated by KDE and the coordinates of the distance space, respectively. The distance between the undetected alleles cannot be obtained. The frequencies of the undetected alleles are generally much lower than those of the detected alleles. Therefore, we ignore the terms for two undetected alleles in the calculation of the approximate nucleotide diversity. For the terms between a detected allele and an undetected allele, frequencies and distance are obtained by extra calculation even when the distance is recorded as “Inf” in the coordinate of the undetected allele. Thus, approximate nucleotide diversity is calculated by ignoring the terms between undetected alleles. Additional calculations to obtain the frequency and distance between detected and undetected alleles are shown in the Supplemental Article.

Implementation

The KDE of allele frequency was implemented using R (version 4.3.2; R Core Team, 2023). This implementation used ape package (version 5.7.1; Paradis & Schliep, 2019) to import Fasta files and depended on matrixStats package (version 1.0.0; https://cran.rstudio.com/web/packages/matrixStats/index.html) to accurately calculate the logarithm of the sum of the exponentials of allele frequencies. Although we prepared the parallelization of the calculation using snow package (version 0.4-4; Tierney, Rossini & Li, 2008), the KDE calculation requires large memory usage for creating space, and the time to copy the memory for parallelization overwhelmed the merits of acceleration by parallelization. Thus, parallelization through multi processes using snow package was not useful.

Experiments

To examine the validity of KDE in estimating genetic diversity, we conducted experiments using simulation and real data. For the simulation experiment, we first prepared fasta files of the sequences using a coalescent simulation. The simulation parameters are listed in Table 1. The coalescent simulation was conducted using ms (Hudson, 2002), but the poor pseudorandom number generator implemented in ms (a linear congruential generator) was replaced by dSFMT, which is a descendant of the Mersenne twister (Matsumoto & Nishimura, 1998). The revised version of ms is available at https://GitHub.com/heavywatal/msutils. The resultant coalescent trees were processed using Seq-Gen (version 1.3.4; Rambaut & Grass, 1997) to obtain fasta sequence files under Generalized time-reversible model (Tavaré, 1986). One sequence per individual was generated, and the sequence length was set to 1,000. We tried to examine the case of a sequence length of 10 using the non-compressed nucleotide sequence space, but the memory was deficient even for a PC with 128 GB RAM, and we abandoned the test. The size of population was set to 100. The size 100 population is a rather rare case in nature, but it helps to observe the effect of population finity on the estimation as it can realize a high rate of samples to the statistical population. The mutation rates per generation were set to 0.01/bp and 0.1/bp. We used these quite high mutation rates to guarantee sufficient mutations in the population; because this coalescent simulation was conducted in a single subpopulation, the sequences rapidly coalesced, resulting in few mutations in the data under normal mutation rates. The sequences produced by the simulation were randomly sampled twenty times using the Mersenne twister pseudo random numbers. The sample sizes were 20, 40, 60, 80 and 100. KDE was applied to these samples under mutation number zero, one and two. We first tried mutation number 3 but abandoned it because of its long calculation time. Mutation number zero is equivalent to not applying KDE. The bandwidth was optimized using both of LSCV and LCV methods.

Table 1 Parameters of the simulation experiment.

Parameter	Value(s) or method(s)	
Population size	100	
Sequence length	1,000	
Mutation rate per generation per base pair	0.1 and 0.01	
Sample size	20, 40, 60, 80 and 100	
Bandwidth optimization method	LSCV and LCV	
Mutation number in KDE	0, 1 and 2	

Using the sample data as well as population data, we calculated the following statistics: Nucleotide diversity, relative bias of nucleotide diversity, accuracy of nucleotide diversity, squared accuracy of nucleotide diversity, concordance rate of allele frequencies between the sample and population as well as their standard deviations. Relative bias and accuracy are defined as the following statistics:

∑i=1n⁡(xi−X)/X

∑i=1n⁡|xi−X|/X,

where n is the sample size, xi is the sample size, and X is the parameter. Concordance rate of the allele frequencies is defined as:

1−∑i=1A⁡|f(ai)−f^(ai)|,

where ai is a coordinate on nucleotide sequence or distant space, A is the size of the space, f(⋅) is the probability mass function of allele frequency of population and f^(⋅) is the estimated probability mass function of allele frequency. The calculation of the concordance rate has two purposes. The first is to measure the effectiveness of the KDE as an estimation of the probability mass function. When the KDE completely estimates the true allele frequency of the population, the concordance rate will be one. For the data with mutation numbers 1 and 2, the differences in the accuracy of nucleotide diversity and concordance rate of allele frequencies were calculated between the positive mutation data and zero mutation data to examine the effectiveness of KDE. The second purpose of calculating the concordance rate is to examine the potential of allele KDE for improved expression of haplotype networks; if allele KDE can provide a better concordance rate including undetected alleles, then it will be possible to construct a haplotype network including undetected alleles which is closer to the true allele frequency than existing ones. Since the undetected alleles around detected alleles in distance space are numerous and probabilistic, the improved haplotype network should be drawn like electron cloud and atomic nuclei in a molecule which correspond to the undetected alleles and detected alleles, respectively, resulting in “haplotype cloud” rather than haplotype network. Genetic diversity measures other than nucleotide diversity, such as expected heterozygosity or Watterson’s theta, were not calculated because they could not be calculated from the allele frequency estimated by KDE on the distant space.

For experiments using real data, we employed a resampling scheme, where real sequence samples were considered as the statistical population. Here, 85 mitochondrial D-loop sequences from Tibetan sheep in Ganba village, China were used (Liu et al., 2016). After retrieving the sequences from GenBank, we aligned them using the online version of MAFFT (ver. 7.511; Kato, Rozewicki & Yamada, 2019) with the default option. Then, we deleted all cites with gaps, which were not able to be co-used with substitutions, resulting in a 1,095-bp alignment length. We randomly resampled 20, 40, 60, 80 and 100% of the sequences using the Mersenne twister. The resampled data were processed in the same way as in the simulation experiments, but calculation for mutation number 2 was not possible due to deficiency of memory even for a machine with 128 GB RAM.

Result

Simulation experiment

The population nucleotide diversities under mutation rates 0.1 and 0.01 were 0.1611368 and 0.0175182, respectively. The squared accuracy of nucleotide diversity showed generally the same tendency as the accuracy. Therefore, we hereafter explain the accuracy of the nucleotide diversity. All the raw data, their graphs and their comparing graphs are shown in Supporting Information. Because all the data are too abundant to show concisely, we representatively show four graphs of the concordance rate and accuracy (Figs. 3–6).

Figure 3 The accuracy of nucleotide diversity under mutation rate 0.01.

The five numbered groups correspond to the following parameter sets: 1. Without KDE. 2. KDE using LSCV and mutation number 1. 3. KDE using LSCV and mutation number 2. 4. KDE using LCV and mutation number 1. 5. KDE using LCV and mutation number 2.

Figure 4 The accuracy of nucleotide diversity under mutation rate 0.1.

The five numbered groups correspond to the following parameter sets: 1. Without KDE. 2. KDE using LSCV and mutation number 1. 3. KDE using LSCV and mutation number 2. 4. KDE using LCV and mutation number 1. 5. KDE using LCV and mutation number 2.

Figure 5 The allele frequency concordance rate under mutation rate 0.01.

The five numbered groups correspond to the following parameter sets: 1. Without KDE. 2. KDE using LSCV and mutation number 1. 3. KDE using LSCV and mutation number 2. 4. KDE using LCV and mutation number 1. 5. KDE using LCV and mutation number 2.

Figure 6 The allele frequency concordance rate under mutation rate 0.1.

The five numbered groups correspond to the following parameter sets: 1. Without KDE. 2. KDE using LSCV and mutation number 1. 3. KDE using LSCV and mutation number 2. 4. KDE using LCV and mutation number 1. 5. KDE using LCV and mutation number 2.

Comparison of mutation number

A mutation number of zero showed the best accuracy for nucleotide diversity (no-KDE in Figs. 3 and 4). Mutation number 2 was the worst in most cases, but mutation number 1 was the worst under a mutation rate of 0.1 and bandwidth optimization by LCV. The bias of nucleotide diversity showed the same trend as nucleotide diversity accuracy. Whereas the bias under mutation number 0 was around 0, those under mutation numbers 1 and 2 were mostly negative.

The concordance rate of allele frequency between the sample and population increased mostly proportional to the sample size (Figs. 5 and 6). Mutation number zero was the highest in the almost all cases. Exceptionally, the parameter set of bandwidth optimization under LSCV and LCV, sample size 20, mutation rate 0.1 and mutation number 1 showed a better concordance rate than that of mutation number 0.

Comparison of bandwidth optimization method

The accuracy of nucleotide diversity under LCV was almost always better than that under LSCV (Figs. 3 and 4). Only when the sample size 20, mutation rate 0.1 and mutation number 1, LSCV showed slightly better accuracy of nucleotide diversity than LCV. The bias of nucleotide diversity was mostly negative, and the bias under LCV was always closer to 0 than that under LSCV.

LCV showed higher concordance rate than LSCV in most cases (Figs. 5 and 6). LSCV showed a higher concordance rate than LCV in two cases: The case of the sample size 20, mutation rate 0.1 and mutation number 1, and the case of the sample size 20, mutation rate 0.1 and mutation number 2.

Comparison of mutation rate

The accuracy of nucleotide diversity under mutation rate 0.01 was always better than that under mutation rate 0.1 when using LSCV. When using LCV, the accuracy of nucleotide diversity under mutation rate 0.1 was better than that under mutation rate 0.01 in the following cases: The case of the sample size 60 under mutation number 1 and case of the sample size 40 and 60 under mutation number 2. The bias of nucleotide diversity was mostly negative. Whereas the bias of mutation rate 0.01 under LSCV was clearly closer to zero, that of under LCV did not show such large difference between the mutation rate 0.1 and 0.01.

Concordance rate under mutation rate 0.01 was almost always higher than that under mutation rate 0.1. Only in the case of the sample size 80 using LCV, the concordance rate under mutation rate 0.1 was higher than that under mutation rate 0.01.

Real data experiment

The population nucleotide diversity was 0.019857. Again, the squared accuracy of nucleotide diversity showed generally the same tendency as the accuracy. Therefore, we hereafter explain the accuracy of nucleotide diversity. All the raw data, their graphs and their comparing graphs are shown in Supporting Information. We representatively show two graphs of the concordance rate and accuracy (Figs. 7 and 8).

Figure 7 The accuracy of nucleotide diversity in the real data experiment.

The three numbered groups correspond to the following parameter sets: 1. Without KDE. 2. KDE using LSCV and mutation number 1. 3. KDE using LCV and mutation number 1. The error bars show the standard deviations.

Figure 8 The allele frequency concordance rate in the real data experiment.

The three numbered groups correspond to the following parameter sets: 1. Without KDE. 2. KDE using LSCV and mutation number 1. 3. KDE using LCV and mutation number 1. The error bars show the standard deviations.

Except when the sampling ratio was 20%, LCV selected almost zero bandwidths, resulting in the almost same values of statistics between LCV and noKDE. Accuracy of the nucleotide diversity under LCV was almost always better than that under LSCV (Fig. 7). LSCV showed better nucleotide diversity accuracy than LCV only when the sampling ratio was 20%. However, even when the sampling ratio was 20%, the value with noKDE was better than that with LSCV. The bias of nucleotide diversity under LSCV was always negative. The bias under LCV was always closer to zero than that under LSCV. LCV and noKDE showed higher concordance rates than LSCV in most cases, but LSCV was higher when the sampling ratio was 20% (Fig. 8).

Discussion

The results of the simulation experiment and real data experiment were mostly concordant. Because mutation number zero showed the best accuracy and bias of nucleotide diversity, KDE on distance space did not improve the estimation of nucleotide diversity. Nucleotide diversity estimated using KDE was mostly negative, probably because the approximate nucleotide diversity ignores the terms between two undetected alleles. If KDE on sequence space is possible, the estimated nucleotide diversity can be improved to some extent. The allele frequency estimated by KDE under LSCV, sample size 20, mutation rate 0.1 and mutation number 1 was better than that estimated without KDE, and this was also the case in the real data experiment under LSCV and 20% sampling ratio. This might imply that the KDE worked better under smaller rate of sample size to the population size. However, it lacks concordance; KDE under LSCV works worse with a larger sample size, and this spoils its general use. This may be due to the multidimensionality of the distance space and/or finity of the population.

KDE is usually applied to a probability density that is not composed of a finite population. Therefore, when KDE is applied to a frequency composed of a finite population, the estimation may be deteriorated, but this is inevitable in allele frequency estimation. Concerning dimensionality, kernel smoothing, which is a more general and comprehensive method than KDE, is known to be suffered from the curse of dimensionality (Blum et al., 2013). As we used distance space instead of sequence space, further dimension reduction may be a remedy for the curse of dimensionality, but it may lead to complete loss of information on individual nucleotide bases. Therefore, even if further dimension reduction improved the estimation of allele frequency, the estimated data could not be used to calculate the existing genetic diversity statistics. Improving the KDE of allele frequencies via further dimension reduction will require the development of new genetic diversity statistics. In contrast to dimension reduction, if KDE on the sequence space were possible, different results might be obtained, and this method does not require new genetic diversity statistics. However, KDE on sequence space requires an unrealistically huge size of memories unless the sequences have only a few bases.

The result comparing LSCV to LCV showed that LCV was better for the estimation of allele frequency and nucleotide diversity. It is reported that LCV is generally more efficient than LSCV except when estimating heavy-tailed distributions (e.g., Wang et al., 2020). In this study, since we severely limited and cut off the tail of distribution by the mutation number, the better performance of LCV is to be expected.

The result comparing the mutation rates did not show large difference in nucleotide diversity bias between the rates when using LCV, but the accuracy of nucleotide diversity was low at mutation rate 0.01. This means that a population with a lower mutation rate is better suited to KDE for the estimation of nucleotide diversity. This trend was also observed in the estimation of allele frequency in the result comparing concordance rates. This result seems to be due to the curse of dimensionality because a higher mutation rate increases the type of observed alleles, which further increases the dimension of the distance space.

Although this study used a simple kernel function which corresponded to the Jukes-Cantor substitution model (Jukes & Cantor, 1969), it is theoretically possible to make the other kernel functions concordant with more complicated substitution models like the general time-reversible model (Tavaré, 1986). In this case, the kernel function changes depending on the sequence. For example, the kernel function will differ when the observed allele is “A” and when it is “G” because the mutation rate of “A” to “T” and “G” to “T” can differ. Such kernel functions can be called anisotropic kernel functions. However, this anisotropy prevents the use of distance space because undetected alleles around a detected allele will no longer have equal frequencies, depending on their distance from the detected allele. Therefore, a new method to compress the sequence space is necessary to use the anisotropic kernel function.

In conclusion, KDE of allele frequency on distance space was not useful to estimate allele frequency and nucleotide diversity in general.

Supplemental Information

Supplemental Information 1 Schematic diagram of definition of the problem. Alleles A and B are shown as black points and they are d distance away.

Concentric circles around the allele A show allele group Ax. In the Fig S1, only A1 to A3. are shown. The problem is how many alleles in Ax correspond to Bd+y(Ax). For example, A1 can include Bd-1(Ax) Bd(Ax) and Bd+1(Ax), but their size, #Bd-1(Ax), #Bd(Ax) and #Bd+1(Ax), requires calculation to obtain.

Supplemental Information 2 Frequency and distance between a detected allele and undetected alleles.

Additional Information and Declarations

Competing Interests

Author Contributions

Data Availability

The authors declare that they have no competing interests.

Satoshi Aoki conceived and designed the experiments, performed the experiments, analyzed the data, prepared figures and/or tables, authored or reviewed drafts of the article, and approved the final draft.

Keita Fukasawa analyzed the data, authored or reviewed drafts of the article, and approved the final draft.

The following information was supplied regarding data availability:

The source codes, experiment data and graphs made in this study are available at OSF: Aoki, Satoshi. 2024. “Kernel Density Estimation of Allele Frequency.” OSF. March 29. https://osf.io/enjw5/.

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
