# Peer review of "Kernel density estimation of allele frequency including undetected alleles"

_PeerJ, doi:10.7717/peerj.17248_

## Round 0.1 · original submission · Major Revisions

Please revise your paper following all of the recommendations of the reviewers, and in particular: 1) Please explain the standard procedure you compare with. 2) Please try to test the method on real data, if possible. 3) Please try to improve the English language. 4) Please explain in detail the substitution model that you used in your simulations. 5) Please try to improve your model by adopting more accurate substitution models, if possible

**Language Note:** The Academic Editor has identified that the English language must be improved. PeerJ can provide language editing services - please contact us at copyediting@peerj.com for pricing (be sure to provide your manuscript number and title). Alternatively, you should make your own arrangements to improve the language quality and provide details in your response letter. – PeerJ Staff

·

Basic reporting

Some of the English could be improved.

Experimental design

While I am familiar with KDE and do work in biology, I am not overly familiar with genetics beyond the basics of allele frequency, etc.. It might be my lack of understanding, but I had a hard time following important aspects of this work. From my reading of the manuscript, the authors were constructing plug-in estimators of various derived parameters based on discrete KDEs, and then comparing those estimators to some standard estimators that were not explained (perhaps because they are so standard). So when the KDE methods were reported to not perform well, I was left asking - compared to what? Even in the figures, the alternative method was just referred to as "noKDE" and "without KDE". This is important for me to understand the statistical value of the work. A major advantage of KDE is that the method is nonparametric. Is there a model assumed by "noKDE"? What is that model? Is it the same model as that of the simulated experiment? If so, there's no reason for that comparison, because KDE is not asymptotically efficient, and so not nearly as good as something like maximum likelihood. An hypothesized advantage stated here for KDE was the inclusion of undetected alleles, but I didn't understand how the "noKDE" method would not account for the basic properties of the data.

Validity of the findings

I can't judge the merits of the comparison without clarification.

Reviewer 2 ·

Basic reporting

I suggest the authors provide the versions of the software and packages.
I also suggest authors provide the R script and relevant code that generates the result, which is beneficial to the reproducibility.

Experimental design

I suggest authors provide some experiments on the selection of kernel function and why the selection fits the problem.
Authors test their methods on simulation data. Is it possible to test the method on real data? I suggest authors provide a case of application of the proposed method.

Validity of the findings

Real data can be more convincing to show the effectiveness of the proposed method. And, it can also show that the problem and demand actually exist.

Reviewer 3 ·

Basic reporting

The authors are to be congratulated on their generally excellent English, but there are a number of errors that could be removed by getting a fluent English speaker to read the manuscript (taking care to choose someone who does actually know some English grammar - there are plenty of English speakers who are hopeless at grammar!). However, there are some points where the text could not be understood:
Line 43: "base composition" not "base component"
Line 54: These sentences are not grammatical, and hard to interpret: "Space on which the probability density function is defined vs. vector space of all possible sequences (nucleotide sequence space). Detected data (sample) vs. detected alleles. Range around the detected data vs. sequences of undetected alleles."
Line 63 "Allele frequency ranges from 0 to 1," Actually, the frequency is an integer between zero and positive infinity eg '47'. The allele proportion ranges from 0 to 1, eg 47/100 = 0.47, where 100 is the total sample of alleles of all types. the population genetics literature is full of this mistake (not made in any other part of science), but there is no reason why you should continue it.
Figure 1: There are no gray bars.

Experimental design

The authors have made a very useful investigation of a promising approach. The negative outcome should not stop publication, but I think there might be a number of ways to improve the outcome, as well as sometimes a bit more definition.
Line 67: Explain what is meant by "define the distance between two bases", eg in what dimensions, etc. This becomes apparent later, but needs to be said now.
Line 76: The reason that the Kernel density estimate did not work well may be because the equation on line 76 is very oversimplified. Suitable models, of increasing complexity, include relatively simple ones like Kimura's two parameter model (transitions A<>G and T<>C have differing probability to transversions A<>T A<>C G<>T G<>C) up to the full GTR Generalised time-reversible model. Also, was this variety of rates included in the coalescent simulation? They should be.
Line 83: "consider that P() and Q() correspond to the bandwidth (smoothing
84 parameter) and the kernel function, respectively." This is crucial to the argument, and needs to be explained and defended, not simply stated.
Line 84: "The mutation rate ... and the number of generations ... are not necessary to conduct KDE." Again, this needs to be explained and defended, not simply stated.
Line 98: "differential of the nucleotide sequence space is difficult to define" This looks like it needs an approach using grad, div and curl.
Line 183-188. What about precision (usually measured as its reverse, coefficient of variation)?
Lines 220-224: I am glad that the manuscript examines the effect of truncation by different values of the 'mutation number', but can there be any general advice about how to choose this?

Validity of the findings

I do not see any reason to doubt the findings given the choices that the authors have made, but as explained above, I think that some of these choices could be altered.

---

## Round 0.2 · accepted · Accept

The authors have satisfactorily addressed the comments by the reviewers.

Reviewer 2 ·

Basic reporting

The issues have been addressed.

Experimental design

The issues have been addressed.

Validity of the findings

The issues have been addressed.